# Undisturbed Soil Pedon under Birch Forest: Characterization of Microbiome in Genetic Horizons

Natalia B. Naumova [1,*] , Ivan P. Belanov [1], Tatiana Y. Alikina [2] and Marsel R. Kabilov [2]

1. Institute of Soil Science and Agrochemistry SB RAS, Lavrentieva 8/2, 630090 Novosibirsk, Russia; belanov@issa-siberia.ru
2. Institute of Chemical Biology and Fundamental Medicine SB RAS, Lavrentieva 8, 630090 Novosibirsk, Russia; alikina@niboch.nsc.ru (T.Y.A.); kabilov@niboch.nsc.ru (M.R.K.)
* Correspondence: naumova@issa-siberia.ru

**Abstract:** Vast areas of land in the forest-steppe of West Siberia are occupied by birch forests, the most common ecosystems there. However, currently, little is known about the microbiome composition in the underlying soil, especially along a sequence of soil genetic horizons. The study aimed at inventorying microbiome in genetic horizons of a typical Phaeozem under undisturbed birch forest in West Siberia. Bacteria and fungi were studied using 16S rRNA genes' and ITS2 amplicon sequencing with Illumina MiSeq. *Proteobacteria* and *Acidobacteria* together accounted for two-thirds of the operational taxonomic units (OTUs) numbers and half of the sequences in each genetic horizon. *Acidobacteria* predominated in eluvial environments, whereas *Proteobacteria*, preferred topsoil. The fungal sequences were dominated by *Ascomycota* and *Basidiomycota* phyla. *Basidiomycota* was the most abundant in the topsoil, whereas *Ascomycota* increased down the soil profile. *Thelephoraceae* family was the most abundant in the A horizon, whereas the *Pyronemataceae* family dominants in the AEl horizon, ultimately prevailing in the subsoil. We conclude that soil genetic horizons shape distinct microbiomes, therefore soil horizontation should be accounted for while studying undisturbed soils. This study, representing the first description of bacterio- and mycobiomes in genetic horizons of the Phaeozem profile, provides a reference for future research.

**Keywords:** ITS DNA diversity; 16S rRNA gene diversity; Phaeozem; soil properties; forest-steppe

## 1. Introduction

Soils are a major component of terrestrial ecosystems. Naturally developed soils are highly structured media representing an integrated system of specific soil environments, i.e., soil genetic horizons, developed by soil genesis under the effect of soil-forming factors (parent material, climate, relief, biota, and time) and sustained by many elementary soil processes. Soil microorganisms, i.e., the organisms less than 5000 $\mu m^3$ [1] in body volume, are important players in many of the elementary soil processes, decomposing plant material, cycling nutrients, controlling pests, supplying water and nutrients to plants, thus contributing to shaping specific sequence of soil genetic horizons in a soil profile. At the same time, specific soil properties within genetic horizons can shape the composition of its microbial assemblage [2]. Forest soils often have sharp vertical stratification, suggesting differential microenvironmental conditions for soil microbiota. Forests occupy almost half of Russia's territory [3]. Forests dominated by white birch form large ecosystems in temperate biomes. Vast areas of land in the forest-steppe of West Siberia are covered by birch forests, the most common and still relatively undisturbed ecosystems there.

Nowadays, soil microbial community census conducted with the use of state-of-the-art metagenomic techniques is an indispensable initial stage for well-integrated and focused ecological research [4]. It is commonly agreed that the taxonomic composition of microbial communities varies substantially between soil environments, but the ecological causes of this variation, including the mechanisms that control the spatial distributions of soil

microbes, remain largely unknown [5]. Although much attention has been focused on studying microbiomes in topsoil layers, subsoil horizons have been studied less often. Consequently, much less is known about the genetic diversity of subsoil microbial communities in particular, and about changes in microbial diversity downwards the undisturbed sequence of naturally formed soil genetic horizons in general. It should be stressed that inventorying microbial diversity in the subsoil horizons, rather than solely in the topsoil, is important for preserving, sustaining, and managing forest ecosystems in the future.

The objectives of the study were (a) to describe the composition and structure of soil bacterial and fungal assemblages residing in genetic horizons of a characteristic Phaeozem pedon under undisturbed birch forest in the south of West Siberia (using Illumina MiSeq sequencing of 16S rRNA genes and internal transcribed spacer, ITS, regions), (b) to reveal the diversity patterns that these assemblages show over the soil genetic horizons' gradient, and (c) to find the relationship between soil horizon characteristics and microbiome diversity changes along the soil profile.

## 2. Materials and Methods

### 2.1. The Study Site

The study site is located in the forest-steppe zone in the south of West Siberia in the Novosibirsk region (55°00' N, 83°04" E). The climate of the region is classified as sharply continental with average maximal temperatures in summer (June–August) ranging 22–26 °C and average precipitation ranging 40–65 mm per month, with 119 days of the frost-free period (https://meteoinfo.ru/en/climate/monthly-climate-means-for-towns-of-russia-temperature-and-precipitation, accessed on 24 February 2021). White birch (*Betula pendula* Roth.) is the main forest-forming tree species (Figure 1a), although some scarce Scots pine trees (*Pinus sylvestris* L.) are also present. Under the canopy of trees, the ground is covered by a rather rich assemblage of herbs and grasses, consisting of 49 species representing 27 families, such as *Caryophyllaceae, Plantaginaceae, Polygonaceae, Hypericaceae, Primulaceae, Lamiaceae, Euphorbiaceae, Rosaceae, Lythraceae, Apiaceae–Ubelliferae*, and others. *Fabaceae* and *Asteraceae* families were the most species-rich, each being represented by seven species. The soil (Figure 1b) is Grey soil, according to the national classification of Russian soils [6], or Grey-Luvic Phaeozem, according to the World Reference Base for soil classification [7].

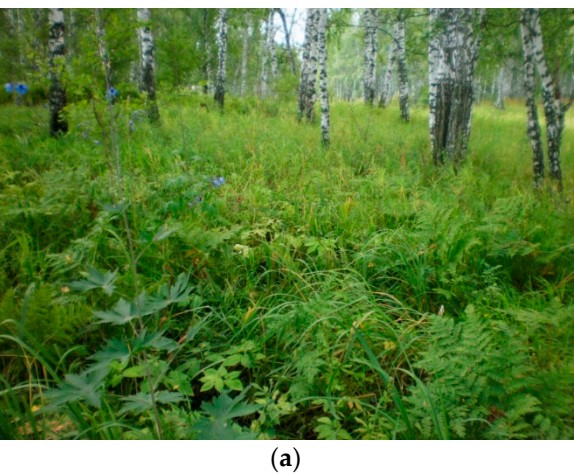

(**a**)

**Figure 1.** *Cont.*

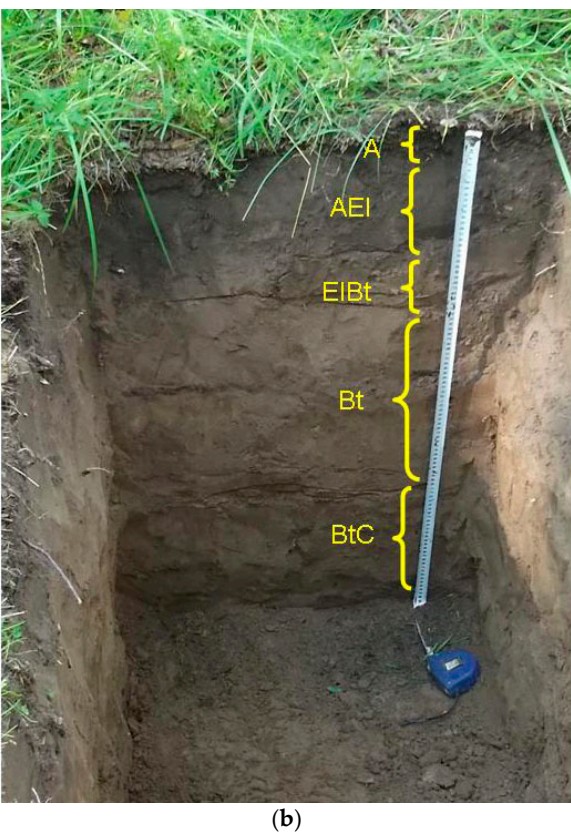

**(b)**

**Figure 1.** The general view of the mature undisturbed birch forest in the forest-steppe in the south of West Siberia (**a**) and the profile of the Gley-Luvic Phaeozem under it (**b**). Letters denote soil genetic horizons.

### 2.2. Soil Sampling and Properties

To study the characteristic soil profile of the landscape, soil samples were taken from a characteristic pedon, pedon being defined as a three-dimensional sample of a body of soil that is 1 m$^2$ at the surface and extends to the bottom of the soil [8]. Soil samples were collected from the three sides of the pedon from the genetic horizons, starting immediately below the litter layer, from the middle part of a horizon. For chemical and microbiological analyses, three individual samples, i.e., replicates, were taken from each horizon, i.e., a total of 15 samples. After sampling, the soil was brought into the laboratory, roots removed, and the soil sieved and thoroughly mixed, and the resulting soil samples were stored at 4 °C for three days prior to chemical analyses. For DNA analyses the aliquots were stored at −20 °C.

### 2.3. Chemical Analyses

Soil organic carbon (SOC) content was estimated by dichromate digestion; soil organic nitrogen content (SON) was determined by the Kjeldahl method; the content of soil labile nutrients (NO$_3^-$, NH$_4^+$, P2O5) and pH (H$_2$O) were measured by standard techniques [9]. Briefly, nitrate content was determined potentiometrically in 0.1% AlKSO$_4$ solution (soil:solution ratio 1:5 *w/v*); ammonium content was measured colourimetrically in 2M KCl extracts (1:10 *w/v*). Available soil P was extracted by 0.03 M K$_2$SO$_4$ (1:5 *w/v*) and determined colorimetrically. Dissolved organic carbon (DOC) content was measured in extracts by dichromate digestion. Soil pH was measured by equilibrating 10 g of field-moist soil with 25 mL of deionized water. Total soil phosphorus was measured by dry digestion [9]. Basal respiration (CO$_2$) was measured by using alkaline traps for CO$_2$ evolved from soil in closed containers [10]. Soil bulk density was calculated as mass/volume ratio after drying a soil core of the known volume at 105 °C for 12 h. The C/N ratio of soil

organic matter was calculated on a molar basis. All analyses were performed in triplicates, and the data were expressed on the oven (105 °C) dry basis (Table 1).

**Table 1.** Properties of soil genetic horizons of the undisturbed Phaeozem under birch forest in West Siberia (means).

| Property | Soil Genetic Horizon | | | | |
|---|---|---|---|---|---|
| | A [¥] | AEl | ElBt | Bt | BtC |
| Depth cm | 0–5 | 5–21 | 21–33 | 33–70 | 70–100 |
| Soil bulk density, g cm$^{-3}$ soil | 0.42 a [#] | 1.14 b | 1.13 b | 1.33 c | 1.47 d |
| pH | 6.5 b | 6.4 a | 6.3 a | 6.5 b | 6.6 b |
| DOC, µg kg$^{-1}$ soil | 56 a | 55 a | 50 a | 43 a | 57 a |
| SOC *, % | 7.51 d | 2.82 c | 1.31 a | 1.41 ab | 1.46 b |
| SON, % | 0.43 d | 0.11 c | 0.05 b | 0.04 a | 0.03 a |
| SCN | 20 a | 31 b | 29 b | 42 c | 55 d |
| NO$_3^-$, mg N kg$^{-1}$ soil | 2.8 c | 1.0 b | 0.8 a | 0.8 a | 0.8 a |
| NH$_4^+$, mg N kg$^{-1}$ soil | 1.4 b | 0.5 a | 1.6 b | 2.2 c | 1.7 b |
| Labile P$_2$O$_5$, mg kg$^{-1}$ soil | 1.2 c | 0.5 a | 0.7 b | 0.7 b | 0.5 a |
| STP, mg·kg$^{-1}$ soil | 85 e | 54 d | 58 c | 53 b | 47 a |
| CO$_2$, µL · hr$^{-1}$ · g$^{-1}$ soil | 11.6 c | 5.1 b | 0.7 a | 0.5 a | 0.9 a |

[¥] Soil genetic horizons: A—humus accumulating horizon, immediately below the litter; AEl—humus-accumulating horizon with features of eluviation; ElBt—eluvial-illuvial horizon; Bt—illuvial horizon; BtC—transitional horizon from Bt to the parent bedrock. [#] Different letters in rows indicate a statistically significant difference at $p \leq 0.05$ (Fisher's LSD test). * Abbreviations used: DOC—dissolved organic C, SOC—soil organic C, SON—soil organic nitrogen, SCN—the C/N ratio of SOC, STP—total soil phosphorus, CO2—soil basal respiration.

### 2.4. DNA Extraction and Sequencing

Total DNA was extracted from 0.40 g of soil using the DNeasy PowerSoil Kit (Qiagen, Germany) as per the manufacturer's instructions. The bead-beating was performed using TissueLyser II (Qiagen, Germany) for 10 min at 30 Hz. The quality of the extracted DNA was assessed by the spectrophotometer NanoDrop ND-1000 (Thermo Fisher Scientific, Wilmington, DE, USA), by agarose gel electrophoresis and pilot PCR. No further purification of the DNA was needed. The amount of the extracted DNA was measured by Qubit (Life Technologies, Carlsbad, CA, USA).

The 16S rRNA genes and ITS2 regions were amplified with the primer pairs V3/V4 343F (5′-CTCCTACGGRRSGCAGCAG-3′) and 806R (5′-GGACTACNVGGGTWTCTAAT-3′), and ITS3_KYO2 (5′¬GATGAAGAACGYAGYRAA-3′) and ITS4 (5′-TCC TCC GCT TAT TGA TAT GC-3′), respectively, combined with Illumina adapter sequences [11]. PCR amplification was performed as described earlier [12]. A total of 200 ng PCR product from each sample was pooled together and purified through MinElute Gel Extraction Kit (Qiagen, Hilden, Germany). The obtained amplicon libraries were sequenced with 2 × 300 bp paired-ends reagents on MiSeq (Illumina Inc., San Diego, CA, USA) in SB RAS Genomics Core Facility (ICBFM SB RAS, Novosibirsk, Russia). The read data reported in this study were submitted to the GenBank under the study accession PRJNA588749.

### 2.5. Bioinformatic Analysis

Raw sequences were analyzed with UPARSE pipeline [13] using Usearch v.11.0.667. The UPARSE pipeline included the merging of paired reads; read quality filtering (-fastq_maxee_rate 0.005); length trimming (remove less 350 nt); merging of identical reads (dereplication); discarding singleton reads; removing chimeras and operational taxonomic unit (OTU) clustering using the UPARSE-OTU algorithm. The OTU sequences were assigned a taxonomy using the SINTAX [14] and 16S RDP training set v.16 [15] or fungi ITS UNITE USEARCH/UTAX v.2018.11.18 [16] as a reference. Taxonomic structure of thus obtained sequence assemblages, i.e., a collection of different species at one site at one time [17], was estimated by the ratio of the number of taxon-specific sequence reads

(archaeal sequences were removed from the data matrix) to the total number of sequence reads, i.e., by the relative abundance of taxa, expressed as a percentage.

The OTUs datasets were analyzed by individual rarefaction (graphs are not shown) with the help of the PAST software [18]: the number of bacterial OTUs detected, reaching a plateau with an increasing number of sequences, showed that the sampling effort (50,000 sequence reads) was close to saturation for all samples, thus being enough to compare diversity [19]. Bacterial OTUs-based $\alpha$-diversity indices were calculated using PAST software [18]. The fungal $\alpha$-diversity indices were calculated for rarefied (33,000 sequence reads) data sets with Usearch v.11.0.667 software.

Statistical analyses (descriptive statistics, correlation analysis, ANOVA, and PCA) were performed by using Statistica v.13.3 and PAST software packages.

## 3. Results

### 3.1. Bacterial Diversity

After quality filtering and chimera removal, a total of 796,099 high-quality 16S gene sequences, generated from the soil samples, were clustered into 4536 different OTUs at 97% sequence identity level, of which the overwhelming majority (4511) was *Bacteria*, the rest representing the *Archaea* domain (removed from further analyses). In total, 23 bacterial phyla were found with 49 identified and 20 non-identified classes.

Most of the total number of bacterial OTUs belonged to the *Proteobacteria* phylum (1126, or 25% of the OTU richness), with *Acidobacteria* and *Actinobacteria* being the second and third most OTU-rich phyla with 11 and 10%, respectively. Notably, many OTUs, were not identified even to the phylum level (29%).

As for relative abundance, the dominance of *Proteobacteria*, *Acidobacteria*, and *Actinobacteria* phyla was much more pronounced as together they accounted for 77% of the total number of sequences in the top horizon, the percentage slightly decreasing downwards (Figure 2a). The relative abundance of bacterial sequence reads that could not be assigned below the domain level, increased with soil depth, accounting in the subsoil horizons for 30% of the total number of sequences.

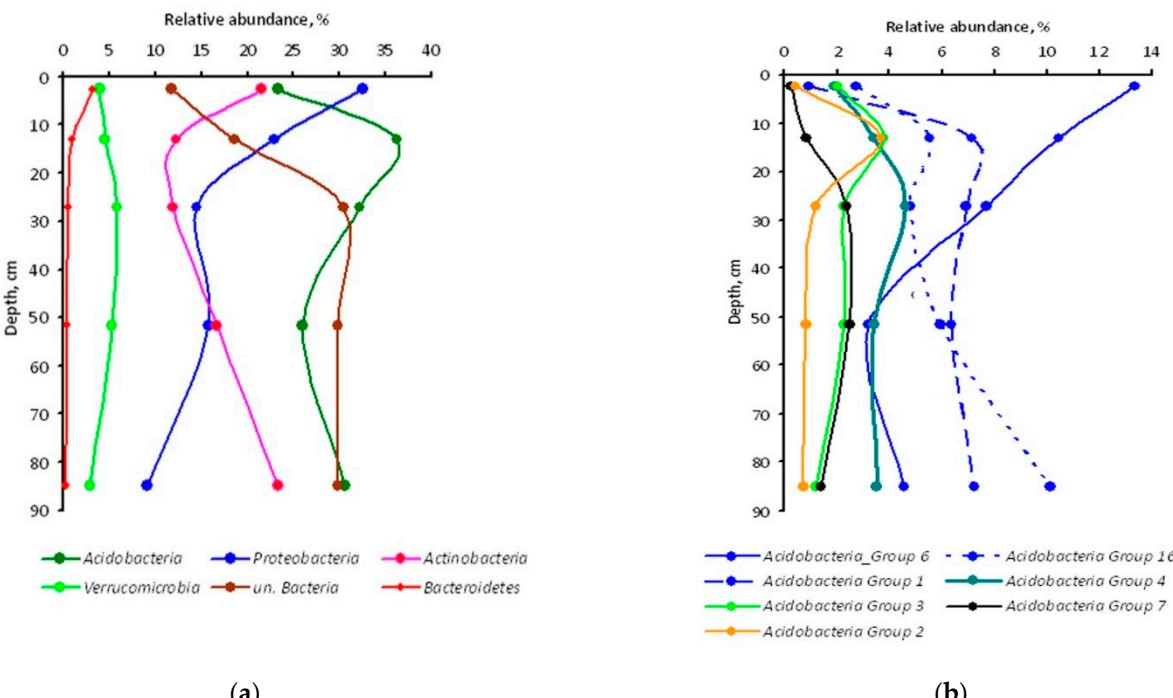

(**a**)　　　　　　　　　　　　　　　　　　　　　(**b**)

**Figure 2.** Relative abundance of bacterial taxa in genetic horizons of the undisturbed Grey-Luvic Phaeozem under birch forest in the south of West Siberia: phyla (**a**) and classes of Acidobacteria phylum (**b**) The markers show the means of three individual soil samples.

About two-thirds of the *Proteobacteria* sequences were represented by the *Alphapro-teobacteria* class in each soil sample, while *Acidobacteria Group 6* class-specific reads contributed 56% to the total number of these phylum-specific reads in the A horizon, decreasing to just 12–15% in the lowest horizons. All other *Acidobacteria* classes in the A horizon were several times less abundant as compared to *Acidobacteria Group 6*, but their abundance increased in eluvial horizons (Figure 2b).

Spearman's correlation coefficients of taxa relative abundance with some soil properties showed specific patterns (Figure 3). The major dominant phylum *Acidobacteria* showed no correlation with soil macronutrient content, correlating negatively with extractable ammonium and labile phosphorus (Figure 3a). The relative abundance of *Proteobacteria* displayed a high positive correlation with soil SOC, SON, STP, basal respiration, and nitrate content, correlating negatively with soil bulk density and C/N ratio of soil organic matter. *Actinobacteria* displayed a positive correlation with soil pH and extractable ammonium, whereas *Verrucomicrobia* showed no statistically significant relationship with the studied soil properties.

At the class level, all four classes of *Proteobacteria* phylum, detected in the study, had the same correlation pattern with soil properties (Figure 3b), while *Acidobacteria* classes showed at least two different correlation patterns (Figure 3c). The dominant *Acidobacteria group 6* correlated positively with SOC, SON, extractable *p* and mineral N, and soil basal respiration, and negatively with soil density and C/N ratio of soil organic matter. The other dominant acidobacterial classes, i.e., *Acidobacteria group 1* and *Acidobacteria group 16*, showing a positive correlation with soil density and C/N ratio, revealed their preference for plant matter that is recalcitrant for decomposition and occurs at lower depths in a soil profile.

The list of the dominant OTUs, i.e., OTUs with a sequence number contributing $\geq 1\%$ of the total number of sequences obtained for a soil sample, collectively for all soil samples amounted to 1929 OTUs, averaging 1165 OTUs per sample. Thus, the majority (57%) of the total number of OTUs were minor or rare members of bacterial assemblages.

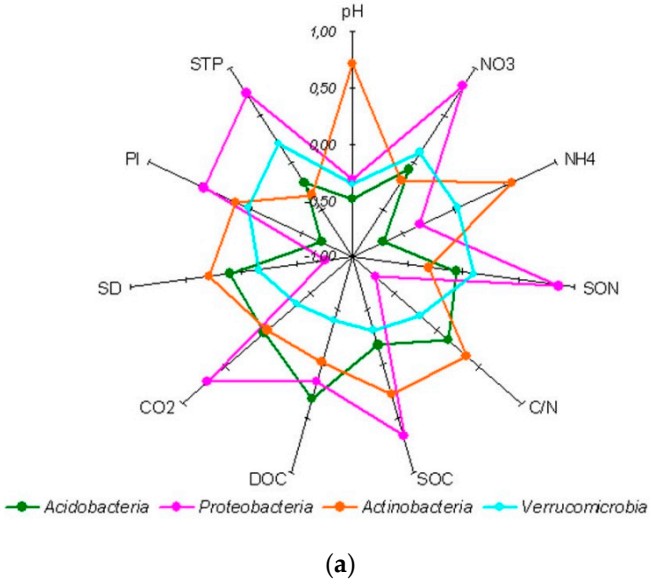

(**a**)

**Figure 3.** *Cont.*

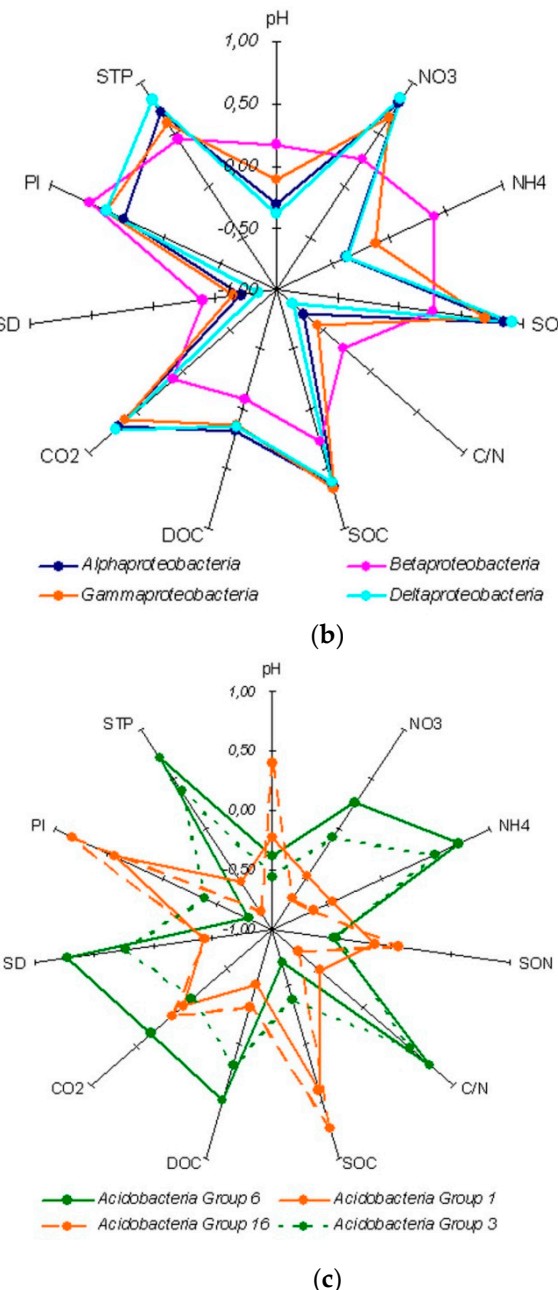

**Figure 3.** Correlation coefficients (Spearman's) between soil properties and relative abundance of dominant bacterial phyla (**a**), Proteobacteria classes (**b**) and the dominant Acidobacteria classes (**c**). The coefficients exceeding │0.51│ are statistically significant at $p \leq 0.05$ level. Abbreviations used: SOC—soil organic carbon content DOC—dissolved organic carbon content, SON—soil organic nitrogen content, STP—soil total phosphorus content, Pl—labile soil phosphorus content, C/N—soil organic matter C/N ratio, CO2—soil basal respiration, SD—soil bulk density.

### 3.2. Fungal Diversity

After quality filtering and chimera removal, a total of 494,400 high-quality fungal ITS gene sequences, generated from the soil samples, were clustered into 679 different OTUs at 97% sequence identity level, of which the overwhelming majority (329, or 48% of the OTU richness) was *Ascomycota*, followed by *Basidiomycota* featuring 135 OTUs (20%), with the other eleven of the identified phyla all together summarily contributing 76 OTUs, or 11% of the total OTU richness in the soil profile. Many OTUs (139, or 20%) represented fungi that could not be classified below the kingdom level.

As for the relative abundance of nucleotide sequences, the dominance of Ascomycota and Basidiomycota phyla was much more pronounced as together they accounted for 90–99% of the total number of sequences in different horizons (Figure 4a), averaging $97 \pm 2\%$ throughout the profile. Notably, the Ascomycota and Basidiomycota relative abundance displayed opposite patterns of soil profile dynamics. Many Ascomycota sequence reads remained unclassified below the phylum level in the ElBt horizon ($12 \pm 4\%$) (Figure 4b), whereas a negligible percentage of Basidiomycota remained unclassified, ranging 0.0–0.7% among the soil horizons.

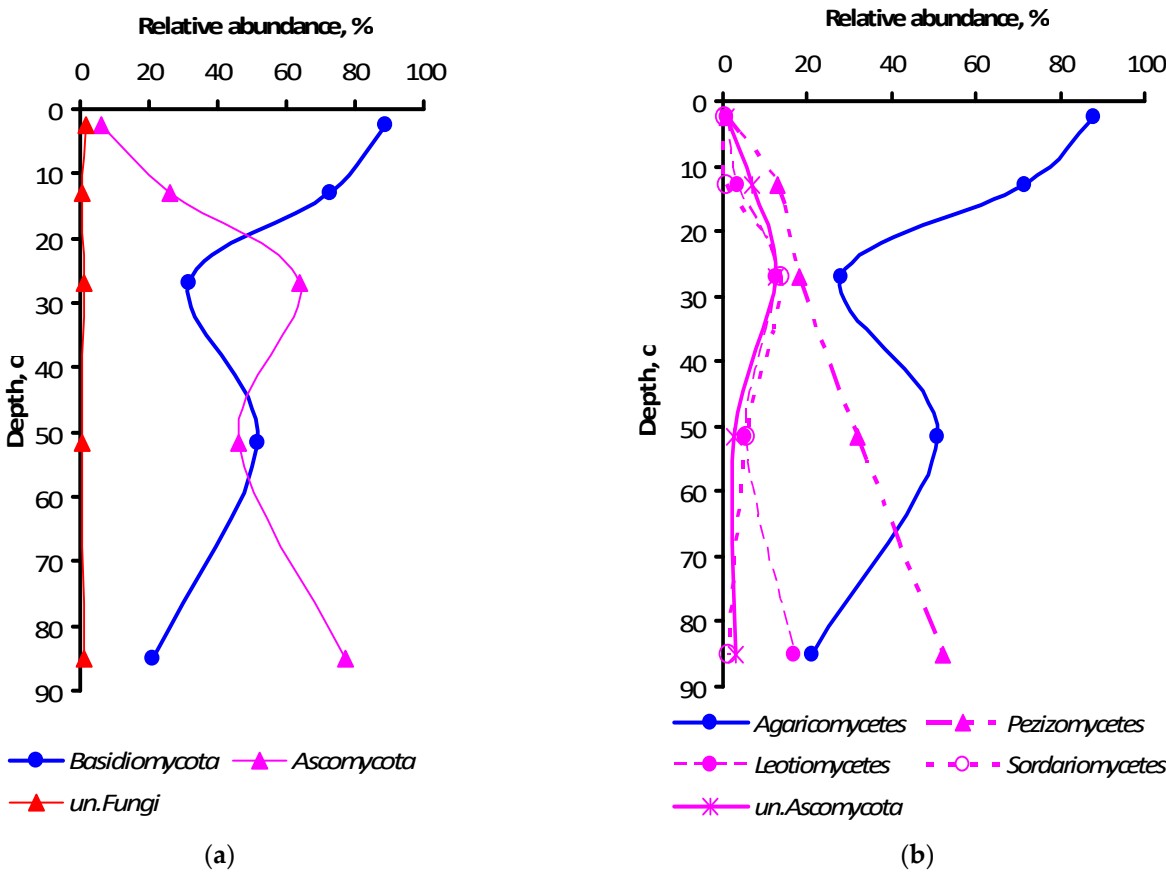

(**a**)  (**b**)

**Figure 4.** Relative abundance of fungal phylum (**a**) and class (**b**) specific ITS amplicon sequences in genetic horizons of the undisturbed Grey-Luvic Phaeozem under birch forest in the south of West Siberia. The markers show the means of three individual soil samples.

The pool of dominant classes for all soil samples numbered 13, ranging from 5 to 11 classes per horizon. The *Agaricomycetes* of *Basidiomycota* was ultimately prevailing in all samples, whereas *Pezizomycetes* of *Ascomycota* ranked second in relative abundance (Figure 4b), except for the A horizon, where it was fifth-abundant with just 1.1%. Further down the taxonomical hierarchy, i.e., at the order level, *Pezizomycetes* were represented solely by *Pezizales*. *Agaricomycetes* were mostly represented by *Telephorales*, *Russulales*, *Sebacinales*, *Polyporale,s* and *Agaricales* orders. As for the family level, *Thelephoraceae* was ultimately predominant in the A horizon ($63\pm\%$), and *Pyronemataceae* was the primary dominant in the three horizons with Bt features, increasing from ElBt ($18\pm\%$) to Bt ($32\pm\%$) and BtC ($52\pm\%$). Overall, the bulked (from all horizons) list of dominating and explicitly identified family-level clusters included 15 families, in addition to those mentioned above featuring *Sebacinaceae*, *Russulaceae*, *Inocybaceae*, *Chaetomiaceae*, *Aspergillaceae*, *Mortierellaceae*, *Umbelopsidaceae*, *Amanitaceae*, *Hyaloscyphaceae*, *Hyaloscyphaceae*, *Cucurbitariaceae*, *Trimorphomycetaceae*, and *Tricholomataceae*, with 10 dominating clusters being unclassified to the family level. At the genus level, *Thelephoraceae* was represented by *Tomentella*; whereas

*Pyronemataceae* was represented by *Wilcoxina*, *Tarzetta*, and *Boubovia* genera, the others being unclassified. Horizon-related differential abundance was revealed for 48 genera at the $p \leq 0.05$ significance level, and for 17 more genera at the $p \leq 0.10$ level, among all of the 14 dominant genera.

At the species level, the overall list of dominants included 39 OTUs, ranging from five in the A horizon to 21 in the ElBt horizon. Both major phyla contributed practically the same number of OTUs into the dominant assemblage, as 19 OTUs were from *Ascomycota* and 18 from *Basidiomycota*, with *Zoopagomycota* and *Mucoromycota* contributing one OTU each. Three dominant OTU-level clusters were identified to a species level (*Lactarius flexuosus*, *Inocybe amblyospora*, and *Umbelopsis dimorpha*). Horizon-related differential abundance was revealed for 122 OTUs at the $p \leq 0.05$ significance level, and for 56 more OTUs at the $p \leq 0.10$ level, among them 20 dominants.

Spearman's correlation coefficients of fungal taxa relative abundance with soil properties displayed specific patterns (Figure 5). The major dominant phylum *Basidiomycota* correlated positively with soil macro- and extractable nutrients content and $CO_2$ evolution, and negatively with C/N stoichiometry and soil bulk density (Figure 5). The relative abundance of *Ascomycota* sequence reads displayed the opposite pattern of correlation with soil properties, which was already indicated by its soil horizon distribution (Figure 4a).

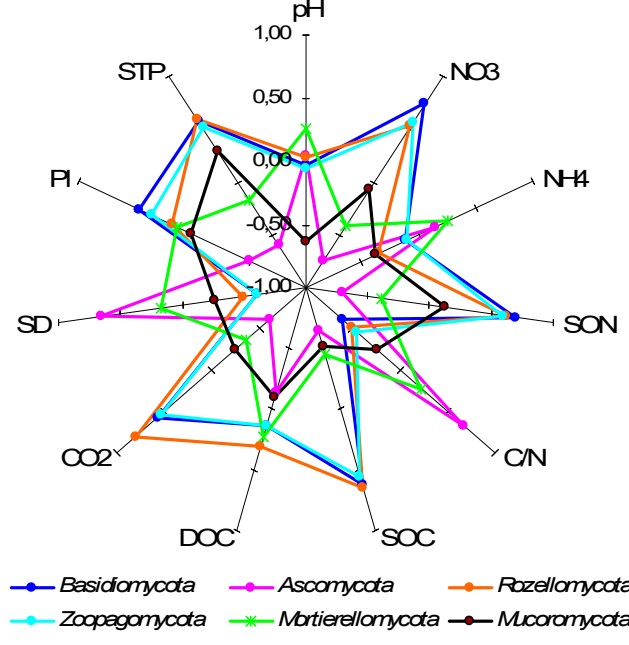

**Figure 5.** Correlation coefficients (Spearman's) between soil properties and relative abundance of some fungal phyla. The coefficients exceeding |0.51| are statistically significant at $p \leq 0.05$ level. Abbreviations used: SOC—soil organic carbon content, DOC—dissolved organic carbon content, SON—soil organic nitrogen content, STP—soil total phosphorus content, Pl—labile soil phosphorus content, C/N—soil organic matter C/N ratio, CO2—soil basal respiration, SD—soil bulk density.

### 3.3. Bacterial and Fungal α-Diversity Indices

Biodiversity indices, calculated on the basis of the total number of bacterial OTUs identified, indicated that bacterial assemblage in the A horizon was more diverse and less dominant (Table 2). Judging by the Chao-1 indices, the number of expected OTUs exceeded the number of the observed OTUs in the top horizon by 12%, i.e., relatively slightly, whereas for the lower horizons the difference was more pronounced, ranging from 34% for the AEl horizon to 20% in the BtC horizon. Dominance was minimal, whereas evenness and equitability were maximal in the upper part of the profile (horizons A, Ael, and ElBt) as compared to the lower horizons (Bt and BtC). The OTUs' richness, Shannon and Chao-1 indices showed a statistically significant ($p \leq 0.05$) positive correlation with soil

macronutrients content, nitrate, and labile phosphorus content and negative correlation with soil specific density and C/N stoichiometry (Figure 6a), the correlation patterns for these three indices being similar. The Simpson's dominance index showed the opposite pattern of significant positive correlation with pH, soil bulk density, and C/N stoichiometry, and negative correlation with soil macronutrients.

**Table 2.** Bacterial operational taxonomic unit (out) biodiversity indices of the Phaeozem genetic horizons under undisturbed birch forest (means).

| Index | Soil Genetic Horizon | | | | |
|---|---|---|---|---|---|
| | A [¥] | AEl | ElBt | Bt | BtC |
| OTUs richness (S) | 1819 d [#] | 1525 c | 1341 bc | 1155 ab | 1016 a |
| Chao-1 | 2541 d | 2234 c | 1823 b | 1607 ab | 1359 a |
| Dominance (D) | 0.007 a | 0.009 ab | 0.007 a | 0.14 b | 0.27 c |
| Berger-Parker | 0.52 b | 0.52 b | 0.34 a | 0.73 c | 0.21 d |
| Equitability | 0.83 d | 0.80 c | 0.82 cd | 0.77 ab | 0.73 a |
| Shannon | 6.3 d | 5.8 c | 5.9 c | 5.5 b | 5.1 a |
| Simpson (1-D) | 0.993 c | 0.991 bc | 0.993 c | 0.999 b | 0.973 a |
| Evenness | 0.18 bc | 0.19 bc | 0.21 c | 0.15 ab | 0.12 a |

[¥] Soil genetic horizons: A—humus accumulating horizon, immediately below the litter; AEl—humus-accumulating horizon with features of eluviation; ElBt—eluvial-illuvial horizon; Bt—illuvial horizon; BtC—transitional horizon from Bt to the parent bedrock. [#] Different letters in rows indicate that the values differ at $p \leq 0.05$ level (Fisher's LSD test).

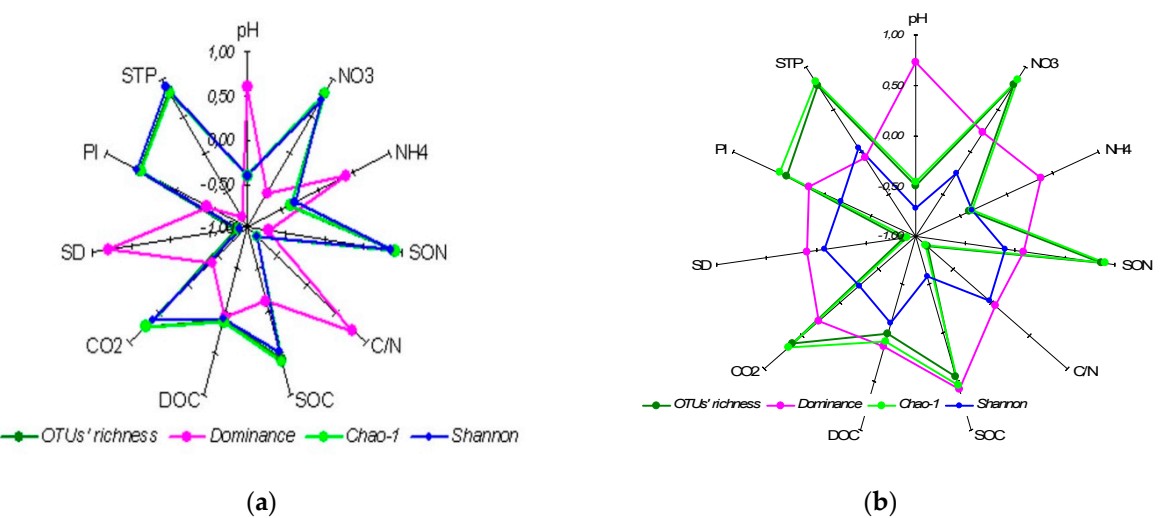

(**a**)  (**b**)

**Figure 6.** Correlation coefficients (Spearman's) between soil properties and bacterial (**a**) and fungal (**b**) $\alpha$-biodiversity indices. The coefficients exceeding |0.51| are statistically significant at $p \leq 0.05$ level. Abbreviations used for soil properties: SOC—soil organic carbon content, DOC—dissolved organic carbon content, SON—soil organic nitrogen content, STP—soil total phosphorus content, Pl—labile soil phosphorus content, C/N—soil organic matter C/N ratio, CO2—soil basal respiration, SD—soil bulk density.

Although fungal OTUs' richness decreased with the depth of soil horizon, most of the $\alpha$-biodiversity indices had either maximal or minimal values in the ElBt horizon (Table 3), altogether reflecting higher $\alpha$-biodiversity in this horizon. Overall, fungal OTUs' richness and Chao-1 indices showed a statistically significant ($p \leq 0.05$) positive correlation with soil macronutrients content and pH (Figure 6b), whereas the Shannon index showed a statistically significant ($p \leq 0.05$) negative correlation with pH and soil organic carbon content. As for Simpson's dominance, the index correlated positively with pH and soil organic carbon content. Notably, the ratio of Chao-1 to the OTUs richness varied with soil

horizon, being maximal in the humus accumulating A horizon (1.38) and minimal in the ElBt horizon (1.06), the difference being statistically significant (*p* = 0.03).

**Table 3.** Fungal OTU biodiversity indices of the Phaeozem genetic horizons under undisturbed birch forest (means).

| Index | Soil Genetic Horizon | | | | |
|---|---|---|---|---|---|
| | A ¥ | AEl | ElBt | Bt | BtC |
| OTUs richness | 191 c [#] | 168 bc | 148 b | 112 ba | 94 a |
| Chao-1 | 260 d | 207 c | 158 b | 131 ab | 106 a |
| Dominance (D) | 0.51 b | 0.14 a | 0.08 a | 0.36 ab | 0.33 ab |
| Berger-Parker | 0.63 b | 0.26 ab | 0.20 a | 0.52 ab | 0.52 ab |
| Equitability | 0.26 a | 0.49 bc | 0.64 c | 0.41 ab | 0.42 ab |
| Shannon | 1.4 a | 2.5 ab | 3.2 b | 1.9 a | 1.9 a |
| Simpson (1-D) | 0.49 a | 0.86 b | 0.92 b | 0.64 ab | 0.67 ab |
| Evenness | 0.17 a | 0.06 b | 0.17 c | 0.07 a | 0.08 ab |

¥ Soil genetic horizons: A—humus accumulating horizon, immediately below the litter; AEl—humus-accumulating horizon with features of eluviation; ElBt – eluvial-illuvial horizon; Bt – illuvial horizon; BtC – transitional horizon from Bt to the parent bedrock. [#] Different letters in rows indicate that the values differ at $p \leq 0.05$ level (Fisher's LSD test).

As expected, the dominance index, both for bacteria and fungi, showed distinctly different correlation patterns with soil properties (Figure 6); whereas in the case of *Bacteria*, actual and potential OTUs' richness and Shannon's indices displayed absolutely similar correlation (Figure 6a) in the case of *Fungi*, not only the dominance index but also Shannon's showed different correlation patterns with basic soil properties (Figure 6b).

The principal component analysis showed clear separation of the A genetic horizon from the subsoil horizons; the separation being more prominent in the case of *Fungi* (Figure 7b,d). Notably, when α-diversity indices were used for PCA, PC1, accounting for 82% for bacteria, was determined by the profile gradient, whereas for fungi, PC1 seemed to be determined by the sample replicates, whereas the downward profile gradient defined PC2, which accounted for 25% of the data total variance.

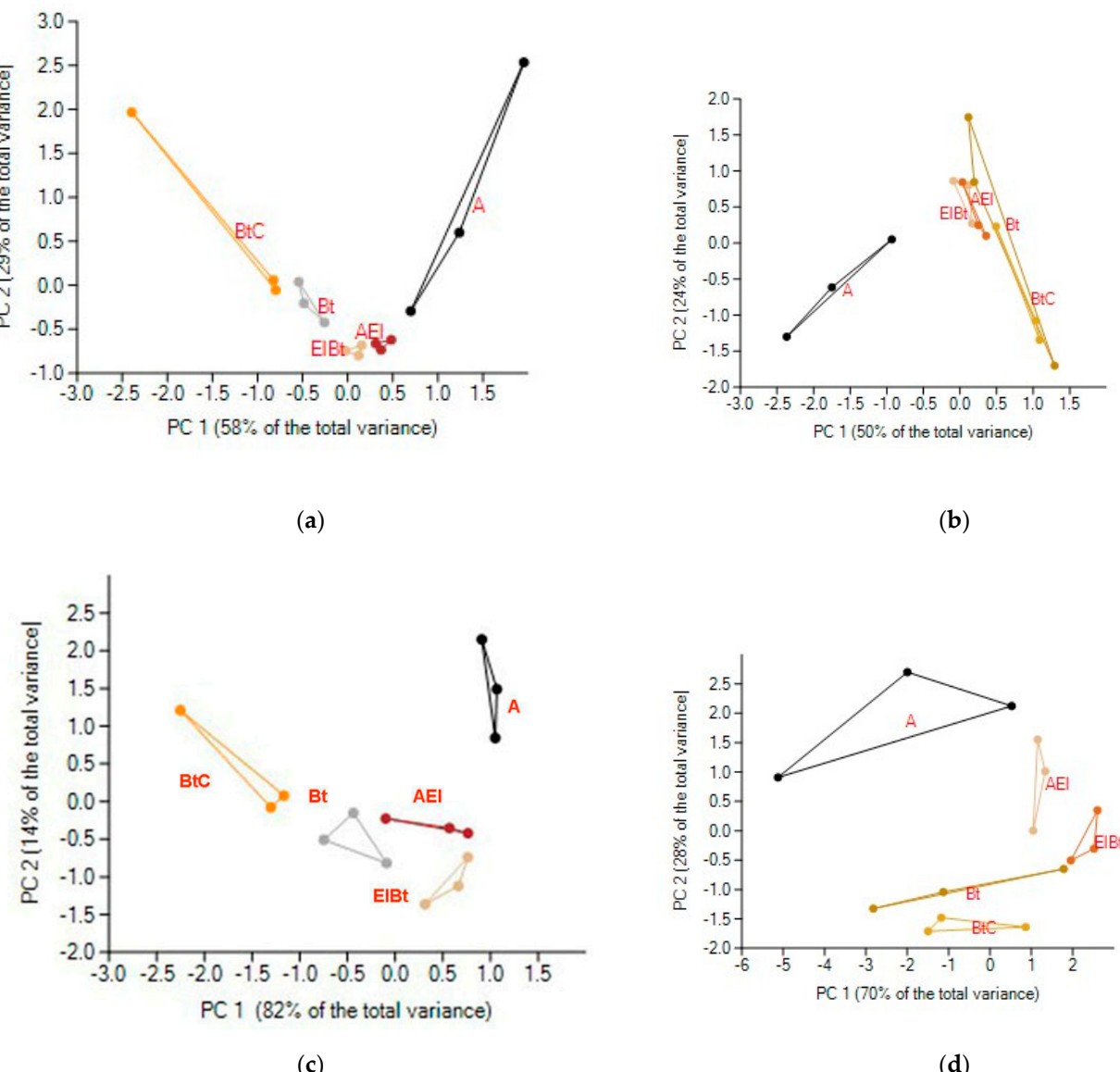

**Figure 7.** Location of the Phaeozem genetic horizons in the plane of the first two principal components of the data matrix with bacterial (**a**) or fungal (**b**) OTU-specific sequence relative abundance bacterial (**c**) or fungal (**d**) α-diversity indices as variables for analysis and soil samples as objects.

## 4. Discussion

### 4.1. Bacterial Assemblages in Soil Genetic Horizons

Overall in our study, 23 bacterial phyla were found. This is much less than 60 phyla reported for some other deciduous forests' topsoils [20,21], with Wei et al. (2018) [21] using the same as our study methodology, i.e., the same primers and sequencing platform. The discrepancy with the latter study might result from the use of different bioinformatic tools for data analysis.

The situation where many bacterial groups remain unidentified is common in soil metagenomic research. These unclassified sequences may represent either novel lineages with no representatives known from cultures, or known bacteria that have not yet been included in the databases, or problems with classification *per se* (see, for example, [22]). In the topsoil horizon of the studied Phaeozem, the percentage of unidentified sequences was close to the values reported for topsoils of other forest ecosystems [23], but down the soil profile, it increased 3-fold. A similar finding was reported for the subsoil of agricultural

soil [24]. Such a situation most likely results from the fact that in soil metagenomic studies, subsoil horizons have been receiving much less attention as compared to the topsoil, and hence their sequences are much less represented in the databases. One cannot help but agree that it is undoubtedly important to look at soil microbial communities in finer detail [25]; we believe that undisturbed subsoil horizons should be included, when possible and feasible, into metagenomic research to get a better picture of niche requirements and properties of taxa and individual microbes. The information on their changes in stratified soil environments will be helpful in drawing hypotheses about their ecological potential and preferences.

Our finding that two bacterial phyla—*Proteobacteria* and *Acidobacteria*—dominated in the organic horizon and throughout the forest soil profile agrees well with the results of other studies in deciduous forest ecosystems [22,26–30]. However, in soil under a secondary natural forest of *Betula albosinensis* in China, *Acidobacteria's* relative abundance was found to reach its maximum (63%) at 40–60 cm soil depth [29], which was not the case in our study where *Acidobacteria* abundance reached its maximum of 37% in the AEl horizon, i.e., 5–21 cm layer (Figure 1b). In contrast to those results, we found no significant correlation between *Acidobacteria* and *Proteobacteria* abundance and dissolved organic carbon. The fact that four major *Proteobacteria* classes showed practically similar correlation patterns with basic soil properties, we tend to interpret as resulting from their prominence in all genetic horizons, coupled with great physiological diversity. Our finding that *Acidobacteria* classes showed a variety of such correlation patterns is, as we see it, confirming their preference for the versatile organic substrates [31], abounding in the undisturbed soil; for instance, we found *Acidobacteria Groups 3* and *6* to be more abundant in soil environments with high C/N ratio, most likely deadwood, relying on available nitrogen supply from other soil sources, whereas *Acidobacteria Groups 1* and *16* tended to prevail in SOM-rich environment, whatever the stoichiometry of the latter.

The dominance of *Acidobacteria Group 6* in the topsoil horizon of the studied Phaeozem complied with similar findings in soil bacterial assemblages under a montane forest ecosystem in Central China [32].

We found that different classes of the *Acidobacteria* phylum had different habitat preferences, as a result displaying different profile distribution patterns and different correlation patterns with soil properties. A positive correlation ($p \leq 0.05$) of the dominant *Acidobacteria Group 6* abundance with soil organic matter, basal respiration, and available nutrients, confirming earlier findings [30], most likely stems from the class's preference for decomposable and diverse organic matter in their environment. The negative correlation of this class's abundance with soil bulk density and soil organic matter C/N ratio also suggests its preference for fresh plant material. Other *Acidobacteria* classes (*Groups 1, 16, 4, 3, 7, 2*) showed maximal abundance in horizons where eluviation processes take place and/or eluvial features are displayed, i.e., AEl and ElBt horizons. There, the slightly lower pH and rather high carbon content (Table 1), as well as potentially some other properties unaccounted for in our study, apparently favor these heterotrophs, capable of utilizing versatile carbon substrates. For example, the second in relative abundance acidobacterial class—*Acidobacteria Group 1*—are able to utilize xylans [33,34], which are the major cell wall constituents in many plants; as well as sugars [35]. Interestingly, this group was found to be the predominant member of *Acidobacteria* in topsoil horizons of forest ecosystems in Germany, while *Acidobacteria Group 6* dominated the phylum in grassland, rather than forest soils [36]. Yet so far, the information about the ecology and physiology of the class has been rather scarce [28,37]. The increase of *Acidobacteria Group 1* and *Group 16* abundance downward the soil profile may indicate that they follow birch roots as the main source of plant material input in the subsoil horizons.

The association of *Actinobacteria* phylum with Bt and BtC horizons is interesting and may result from the input of tree root litter and its utilization by *Actinobacteria*, as they are well-known deadwood decomposers [38]; and/or from the increased presence of phosphorous-solubilizing *Actinobacteria* representatives, such as *Frankia* spp. and others, etc.

The presence of *Actinobacteria* deep in subsoils in *Robinia* forest ecosystems was reported earlier [30].

We found that some *Bradyrhizobiaceae* play an important role in shaping the structure of microbial assemblages of different soil genetic horizons in undisturbed Phaeozem under birch forest. Previously, the non-symbiotic *Bradyrhizobium* ecotypes were found to dominate the North American soils under different forests [39], where nearly every soil sample was dominated by one OTU affiliated with the genus *Bradyrhizobium*, sometimes up to 30%. Previously, we also found one *Bradyrhizobium* sp. OTU dominating the rhizosphere soil and bulk soil under Korean pine [40,41]; and one *Rhizobium* OTU ranking second most dominant both in the rhizosphere and bulk soil, although their dominance did not exceed 4–5% of the total number of sequences in a sample. Summarily, such a prominent presence of non-symbiotic diazotrophs in forest soils underscores their important role in supplying plants with available nitrogen.

Our finding of the somewhat increased *Nitrospirae* relative abundance down the soil profile is consistent with their chemolithoautotrophic nature and the decrease in available nutrients and SOC down the soil profile. In grassland soils (Stagnosol and Cambisol), *Nitrospirae* was found to be much more abundant in B horizon as compared to the A horizon [42]; and recently, the phylum was shown to be one of the major phyla throughout the floodplain soil profile except for the surface organic matter rich layer [43]. Apparently *Nitrospirae* may play an important role in nitrogen transformation in the subsoil.

The α-diversity indices calculated for bacterial 16S rRNA gene sequence assemblages in A horizon were higher (observed OTUs, Chao-1) or somewhat lower (Shannon) than the values reported for the topsoil under a *Betula albosinensis* forest [29]. The Chao-1 and Shannon indices were significantly positively correlated with soil organic carbon content, i.e., with topsoil environment with high and chemically diverse plant material input from grasses, herbs, shrubs, and trees. The revealed positive correlation of bacterial dominance index with soil bulk density and broader organic matter C/N ratio may be indicative of the selective pressure exerted on bacterial assemblage structure by specific environment in the subsoil genetic horizons, i.e., more recalcitrant plant material (mostly wood, or deadwood) and higher density.

The observed horizon-related variation in bacterial assemblage composition and structure confirmed the chemical, physical and biological specificity of soil genetic horizons as habitats for bacteria. Some interesting correlations, found in our study, will be helpful to shape future research to get a better insight into the ecological causes of soil bacteriobiome variation within a soil profile.

### 4.2. Fungal Assemblages in Soil Genetic Horizons

Our results showed that *Basidiomycota* phylum was the ultimate dominant in the undisturbed Phaeozem topsoil genetic horizons, followed by *Ascomycota*. This finding is consistent with earlier results [44], as well as with results about fungal assemblages on leaves *Betula pendula* [45], i.e., the forest-forming tree species and the main edification in our study. The distinctly opposite patterns of soil profile dynamics showed by *Ascomycota* and *Basidiomycota* phyla (with *Ascomycota* increasing and *Basidiomycota* decreasing down the soil profile) prove their differential ecological niche requirements.

The highest observed and expected fungal OTUs' richness in the topsoil A and AEl horizons apparently resulted from the high content and diversity both of SOM and plant material, as 51 plant species grow on the study site. However, in these horizons, the gap between expected and observed OTUs' richness was rather big, indicating a high occurrence of singletons and doubletons [46], i.e., a substantial number of OTUs, represented by one or two sequence reads. This finding suggests that diverse soil conditions in the topsoil horizons are beneficial for many fungi, providing diverse microzones sustaining rare species. As for the Phaeozem subsoil horizons, the number of expected fungal OTUs was only slightly (by 6% in ElBt horizon) exceeding the number of the observed OTUs. The negative correlation of fungal Shannon index with soil organic carbon, and hence with

depth, might be attributed to the prevalence of birch-derived plant material input into the subsoil, which decreases fungal biodiversity due to favoring a sub assemblage of (mostly birch-specific) mycorrhiza and deadwood decomposers.

The most abundant in the A horizon *Thelephoraceae* family (*Thelephorales/Agaricomycetes/Basidiomycota*) was represented by *Tomentella* genus, i.e., a well-known ectomycorrhizal fungi [47]. Another mycorrhizal family, *Sebacinaceae*, also being one of the major dominants in the A horizon, ultimately prevailed in the AEl one, with *Sebacina* sp. accounting for almost a quarter of the total number of sequence reads. The *Pyronemataceae* family, shown to be one of the major dominants in the AEl horizon and the first-ranked in abundance in the lower horizons, are very diverse ecophysiologically [48]; in this study, the family was represented by three explicitly classified genera, namely *Wilcoxina, Tarzetta*, and *Boubovia*, all being common mycorrhizal fungi [49]. However, their dominance was not associated with subsoil environments before. The revealed difference in the mycorrhizal fungal assemblages between the top- and subsoil horizons is most likely due to the difference in the species composition of the belowground living phytomass in the respective horizons, which is apparently greater in the topsoil than in the subsoil. Since assessing plant biodiversity in the belowground layers is a challenging task, that is why so far there is information available only about the link between the aboveground plant biodiversity and soil fungal biodiversity (see, for example, [50]).

A positive correlation between *Basidiomycota*, total soil nutrients content, and basal respiratory activity is indicative of their major contribution to soil microbial biomass and activity.

We want to stress that it is difficult to find the microbiome data with which to compare our results because most of the researchers collected samples from soil layers, rather than soil horizons even in the undisturbed forest ecosystems, and proceeded to compare such soils across different biomes [50], thus ignoring the fact that soil horizons represent specific environmental entities of varying thickness [51] and may be mixed by sampling a layer in centimeters. A sample, taken from the 0–10 cm layer of the mineral topsoil under undisturbed temperate forest may, as it would have been in our case, represent a mixture of two soil genetic horizons with all the consequences for microbial diversity assessment. Of course, one can always collect soil samples from incremental layers, but in the case of undisturbed soils, the sampling increments should fall within a horizon to make any comparisons methodologically relevant.

## 5. Conclusions

To the best of our knowledge, here we present the first study describing changes in the soil bacteriobiome and mycobiome diversity along the sequence of genetic horizons of the undisturbed Phaeozem pedon under birch forest, a widely spread ecosystem throughout the forest-steppe zone in the south of West Siberia, using high-throughput sequencing techniques. We believe the obtained information will help to focus further on more extensive research on soil microbiome relative to the soil environment and also serve as a reference for revealing changes that are likely to occur in the future due to climate warming, human activities, different land use, or other factors. It is surprising that, despite the common agreement that "the study of soil profiles is essential to the study of soil as a natural body" [51] p. 4, there are (a) the scarcity of the metagenomic exploration of the subsoil horizons, resulting in much greater portions of unidentified members of the microbiome in case subsoils are studied; and (b) neglect for sampling soil genetic horizons while studying the undisturbed soils. We conclude that soil genetic horizons shape distinct microbiomes, therefore, soil horizontation should be accounted for while comparing undisturbed soils latitudinally. As an important soil characteristic, soil horizontation deserves more comprehensive research by combined metagenomic and general microbiological techniques, which will most likely find novel taxa and/or novel ecotypes of the already well-characterized taxa from other habitats, thus expanding our

understanding of how the system of soil genetic horizons shape and sustain microbial diversity, as well as contributing to sylvicultural landscape engineering.

**Author Contributions:** Conceptualization, N.B.N. and I.P.B.; methodology, N.B.N.; software, M.R.K.; validation, I.P.B., T.Y.A. and M.R.K.; formal analysis, M.R.K.; investigation, I.P.B.; resources, M.R.K.; data curation, N.B.N.; writing—original draft preparation, N.B.N.; writing—review and editing, M.R.K.; visualization, T.Y.A.; supervision, I.P.B.; project administration, I.P.B.; funding acquisition, I.P.B. and M.R.K. All authors have read and agreed to the published version of the manuscript.

**Funding:** This research was funded by the MINISTRY OF SCIENCE AND HIGHER EDUCATION OF THE RUSSIAN FEDERATION (MSHE RF), projects no. AAAA-A17-117020210021-7 and AAAA-A17-117030110078-1. The APC was funded by the MSHE RF.

**Institutional Review Board Statement:** Not applicable.

**Informed Consent Statement:** Not applicable.

**Data Availability Statement:** The read data reported in this study were submitted to the GenBank under the study accession PRJNA588749, available at https://www.ncbi.nlm.nih.gov/bioproject/PRJNA588749/ (accessed on 29 December 2020).

**Acknowledgments:** The authors thank Natalia V. Sheremet (Central Siberian Botanical Garden, SB RAS, Novosibirsk, Russia) for describing the composition of plant assemblages at the study site.

**Conflicts of Interest:** The authors declare no conflict of interest. The funders had no role in the design of the study; in the collection, analyses, or interpretation of data; in the writing of the manuscript, or in the decision to publish the results.

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
