# Peer review of "Undisturbed Soil Pedon under Birch Forest: Characterization of Microbiome in Genetic Horizons"

_soilsystems, doi:10.3390/soilsystems5010014_

Round 1

Reviewer 1 Report

General view  

This manuscript describes the study on the soil microbiome (diversity and abundance on ITS and 16S genes basis) by genetic soil horizons of one soil pit (15 soil samples with replicates included) in a birch forest in West Siberia. The authors analysed the vertical distribution of soil microbial diversity and how this links to other soil physicochemical characteristics per genetic horizon. This study is a report about the soil microbial distribution for this soil pit which can be representative for the soil serie and as such I would recommend changing the title to say so – I don’t think this pit would represent the entire area for this forest. To represent the entire soil area for this forest might require a deeper and broader soil sampling. I also recommend deeper research and understanding about diversity indices and the differences in the information they expose. The manuscript requires more details such as comparisons and references to other studies about the microbial vertical distribution even when they don’t refer to soil genetic horizons since this is the main innovation. There are many more studies on the horizontal distribution as the once mentioned in the discussion part – Do these communities represent similar pattern in-depth?. The manuscript can be improved after major and minor corrections as described below. For example, some technical concepts or phrases need revision to be aligned with a formal scientific language (e.g. soil bulk density, others, the use of ‘quite a lot’).

Abstract

Line 22 – misspelling ‘horizontation’

I./ Introduction. There is a lack of references in some sentences and, hopefully, a few of the used ones can be updated to more recent years – e.g. line 47. It needs more information such as findings of soil microbiomes in other similar studies even if they are not in genetic horizons. I suggest beginning the introduction with a sentence that introduces the impact or importance of this study in term of the soil microbiome in the forest rather than describing the Russian forest itself.- Other suggestion as following:

Line 34 – perhaps a reference to soil formation factor can complete the idea.

Line 39 – soil environment is too broad - I would better suggest specific soil properties/characteristics.

Line 40– 41 check for references

Line 49– 51 check for references

Line 34 – specify in () at least one soil characteristic

Line 79–80  pedon definition is not needed but can be added as a footnote if this aligns with this journal requirements.

II./Material and Methods. Some details must be added as if someone needs to replicate the study. For instance, the universal primers used should describe their sequences. Did you quantify DNA and PCR products concentration and quality? Is this a dual paired-end protocol? More details should be described to clarify the bioinformatics settings for diversity analyses. Indeed, some basic information is not available, as the number of high-quality sequences. Perhaps, the rarefaction curves as supplementary materials. Please consider that this should not be too extensive.

Line 86 – remove (+) symbol before 4oC

Line 98 – Error! Message

Line 98 – basal respiration method should be specified even though not necessarily describes

Line 99 – it is confusing the so-called ‘specific density’ for soil bulk density along with the manuscript. Please check.

Line 131 – thus or this?

Line 133 – Can be the archaea data be added in supplementary materials? They are also part of the microbiome and could add robustness to your study.

Line 138 – the depth of sampling effort should be specified.

III./Results.

Line 148 – information specified in () is not needed.

Line 152 – not identified OTUs depends on the taxonomic system. Do you mean they might represent Nobel organisms? Be more specific.  

Line 170 – 174 extractable ammonium, nitrate etc.-  Please check on using acronyms for properties as they will appear in the table along with the manuscript. -e.g basal respiration called CO2 evolution, etc. They should be uniform and avoid confusions – e.g. soil bulk density and soil specific density etc.  C/N ratio of soil organic matter also called C/N stoichiometry sure???

Line 182 – 183 I don’t understand the connection to their preferences for recalcitrant plant matter?? Any reference?

Line 188 – usually other studies set up a threshold to assess rare communities. Perhaps, this could help to make this study more robust.

Line 189 – section 3.2 is more about Fungi abundance and distribution than diversity. Most usually diversity refers to diversity indices.

Line 194 – check all informal language along with the manuscript – e.g. quite a lot

Line 199 – Ascomycota vs Basidiomycota both with cursive??

IV./Discussion.  Discussion can be shortened. Most important to include is the contribution of this study to general knowledge. What the purpose of this kind of studies and what would be the difference with before and after your study.   

Line 284 – 287  other forest but in the same soil type/climatic conditions? I think is not appropriate to assume that the discrepancy is due to bioinformatic tools.  

Line 296 – 298 then you should include unclassified OTUs in diversity indices calculations to understand microbiome diversity as well as archaea group. Other options are the use sequence variants analysis instead of OTUs.

Line 310 – 320 maybe some references to pH preferences should be included

Line 371 – OTUs isn't a diversity index but ‘Observed OTUs’.

Line 398 – writing sounds weird  

Line 398 – 404 reference?

Line 428 – Error!

Line 431-435 Interesting discussion. Check how this relates to phenosoils vs genosoils https://www.sciencedirect.com/science/article/pii/S0016706117322401

IV./Conclusion. See contribution from these studies in undisturbed soils to assess losses in soil biodiversity by climate change and other human activities…

Line 444 – Error!

Line 449 – misspelling

English and grammar should be improved throughout the entire manuscript. e.g. formal scientific language.

Author Response

Response to Reviewer 1

Abstract

Point 1

Line 22 – misspelling ‘horizontation’

Response 1

We checked it and dare say we do not see any misspelling there. As you know, “horizontation” means 1) the structure of a natural body in a sequence of horizons, and 2) the methodology to sample such object and interpret the results. The term is used in this context in studies related to soil science. Here are some recent examples.  “For that purpose, other soil properties such as texture, water‐holding capacity, permeability, horizontation, or the extension of the modeled depth to the depth of bedrock may be of interest” (Rentschler et al., 2020, https://acsess.onlinelibrary.wiley.com/doi/full/10.1002/vzj2.20062).

Another example: “Differences in Fe stocks, vertical distribution of Fed, Feo and Fep and horizontation of equally old soils indicated an influence of microclimate on pedogenesis determined by slope aspect.” (Pollmann et al., 2020, abstract, https://www.sciencedirect.com/science/article/abs/pii/S2352009419302457)

Introduction

Point 2

There is a lack of references in some sentences and, hopefully, a few of the used ones can be updated to more recent years – e.g. line 47.

Response 2

We are afraid we cannot understand the need to update the references to more recent ones, in the case the results and conclusions of the cited ones are perfectly justified, solid, robust, actual and pertaining to our study. Personally, we would not want our published materials to be kicked out from referenced works only on the grounds of the more recent date of publication. We firmly believe that scientific citation should be based solely on the general merit and specific relevance of publications. Yet we updated one reference (highlighted in turquoise in the revised version). Besides, we tried to keep the introduction brief and to the point, and the reference list short by not supporting widely acknowledged, as we see them, statements.

Point 3

I suggest beginning the introduction with a sentence that introduces the impact or importance of this study in term of the soil microbiome in the forest…

Response 3

We rearranged two paragraphs in the Introduction section, starting it with soil since we submitted the manuscript for Soil Systems, and putting “forests” somewhat below. Since we did not rewrite anything, we did not highlight this rearrangement.

Point 4

Line 34 – perhaps a reference to soil formation factor can complete the idea.

Response 4

We substituted the phrase “developed due to soil genesis” with “developed by soil genesis under the effect of soil-forming factors (parent material, climate, relief, biota and time)”; the change is highlighted in turquoise in the revised version.

Point 5

Line 39 – soil environment is too broad - I would better suggest specific soil properties/characteristics.

Response 5

We substituted “environment” with “properties”; the change is highlighted in turquoise in the revised version.

Point 6

Line 40– 41 check for references

Response 6

By “sharp vertical stratification” we meant visually very drastic colour and texture differences between soil horizons, like in Podzols, for instance. Those are textbook profiles, broadly known to soil scientists and hence requiring no special references. However, we agree that we were too hasty to attribute the formation of such stratification only to “the decomposition of plant material and other organic matter and the weathering of the parent bedrock” which might require some references in support. So we substituted “resulting from the decomposition of plant material and other organic matter and the weathering of the parent bedrock” with “suggesting differential microenvironmental conditions for soil microbiota”.

Point 7

Line 49– 51 check for references

Response 7

Before submitting the manuscript to Soil Systems, we tried to find the relevant references and failed. If some soil horizons were studied, those are most often the studies of agricultural soils, not undisturbed natural ones, and mostly topsoil, not subsoil horizons. Most often, when the authors write ”soil profile”, the latter means just the depth/thickness of a soil layer and incremental sampling in layers of certain thickness in cm, without following a sequence of soil genetic horizons and mostly shallow, i.e.not reaching into the subsoil.

Point 8

Line 34 – specify in () at least one soil characteristic

Response 8

We substituted the phrase “developed due to soil genesis” with “developed by soil genesis under the effect of soil-forming factors (parent material, climate, relief, biota and time)”; the change is highlighted in turquoise in the revised version.

Material and Methods

Point 9

Line 79–80 pedon definition is not needed but can be added as a footnote if this aligns with this journal requirements.

Response 9

We believe that the definition of a pedon might not be well known to every reader, as, as it was the main object of sampling, the definition is pertinent to the Section.

Point 10

Some details must be added as if someone needs to replicate the study. For instance, the universal primers used should describe their sequences.

Response 10

We inserted the required sequences; the change is highlighted in turquoise in the revised version.

Point 11

Did you quantify DNA and PCR products concentration and quality?

Response 11

We substituted the phrase “The quality of the DNA was assessed using agarose gel electrophoresis”  (Lines 113-114 in the reviewed version of the manuscript) with “The quality of the extracted DNA was assessed by the spectrophotomer NanoDrop ND-1000 (Thermo Fisher, USA), by agarose gel electrophoresis and pilot PCR.  No further purification of the DNA was needed. The amount of the extracted DNA was measured by Qubit (Thermo Fisher, USA)”. The change is highlighted in turquoise in the revised version.

Point 12

Is this a dual paired-end protocol?

Response 12

Yes, it  was: “The obtained amplicon libraries were sequenced with 2x300 bp paired-ends reagents on…” (Lines 119-120  in the reviewed version of the manuscript).

Point 13

More details should be described to clarify the bioinformatics settings for diversity analyses.

Response 13

We appropriately rearranged and supplemented the text: “ … the number of bacterial OTUs detected, reaching plateau with increasing number of sequences, showed that the sampling effort (50,000 sequence reads)… Bacterial OTUs-based α-diversity indices were calculated using PAST software [20]. The fungal α-diversity indices were calculated for a rarefied (33,000 sequence reads) data sets with Usearch v.11.0.667 software”; the changed text and the added reference are highlighted in turquoise in the revised version.

Point 14

Indeed, some basic information is not available, as the number of high-quality sequences.

Response 14

We added “796,099 high-quality 16S gene sequences, generated from the soil samples, were clustered into…” and “494,400 high-quality fungal ITS gene sequences, generated from the soil samples”; the additions are highlighted in turquoise in the revised version.

Point 15

Perhaps, the rarefaction curves as supplementary materials.

Response 15

We are afraid there are already many figures in the manuscript; and distracting readers with some more might be counterproductive.

Point 16

Line 86 – remove (+) symbol before 4oC

Response 16

The (+) symbol removed; the change is highlighted in turquoise in the revised version.

Point 17

Line 98 – Error! Message

Response 17

The “Error!” message was substituted with a correct reference number; the change is highlighted in turquoise in the revised version.

Point 18

Line 98 – basal respiration method should be specified even though not necessarily described

Response 18

We added the phrase “by using alkaline traps for CO2 evolved from soil in closed containers “; the addition is highlighted in turquoise in the revised version.

Point 19

Line 99 – it is confusing the so-called ‘specific density’ for soil bulk density along with the manuscript. Please check.

Response 19

We substituted “specific soil density” with “soil bulk density” through the manuscript; the changes are highlighted in turquoise in the revised version.

Point 20

Line 131 – thus or this?

Response 20

Assemblages obtained how? – Thus obtained assemblages.  

Point 21

Line 133 – Can be the archaea data be added in supplementary materials? They are also part of the microbiome and could add robustness to your study.

Response 21

Unfortunately, we strongly doubt that those very few archaeal sequences that by chance were “fished” out and amplified with universal bacterial primers, resulting in 25 OTUs of the total 4536,  would add any robustness to the study. The concept of robustness of a microbiome study can briefly be defined as based on the use of different methods on the same experimental system (Schloss, 2018). So we do not see as how adding some very incomplete information about Archaea may be helpful in this respect. You are absolutely right that Archaea, undoubtedly, deserve special attention, and, as we have the DNA extracts in store, we may think to proceed with special primers for Archaea.

Reference

Schloss P. D. (2018). Identifying and Overcoming Threats to Reproducibility, Replicability, Robustness, and Generalizability in Microbiome Research. mBio, 9(3), e00525-18. https://doi.org/10.1128/mBio.00525-18

Point 22

Line 138 – the depth of sampling effort should be specified.

Response 22

We specified it  by adding “…the sampling effort (50,000 sequence reads)” for bacteria and “The fungal α-diversity indices were calculated for a rarefied (33,000 sequence reads) data sets with Usearch v.11.0.667 software”; the additions are highlighted in turquoise in the revised version.

Results

Point 23

Line 148 – information specified in () is not needed.

Response 23

The information in () removed; the change is highlighted in turquoise in the revised version.

Point 24

Line 152 – not identified OTUs depends on the taxonomic system. Do you mean they might represent Novel organisms? Be more specific.

Response 24

Since it is in the Results Section, we just state the finding, but then address it in the Discussion: “The situation that many bacterial groups remain unidentified is common in soil metagenomic research.  These unclassified sequences may represent either novel lineages with no representatives known from cultures, or known bacteria which have not yet been included into databases, or problems with classification per se (see, for example, [22]).”  (Lines 289–293 of the manuscript pdf-file you reviewed)

Point 25

Line 170 – 174 extractable ammonium, nitrate etc.-  Please check on using acronyms for properties as they will appear in the table along with the manuscript. -e.g basal respiration called CO2 evolution, etc. They should be uniform and avoid confusions – e.g. soil bulk density and soil specific density etc.  C/N ratio of soil organic matter also called C/N stoichiometry sure???

Response 25

We checked and made the acronyms used consistent; the change is highlighted in turquoise in the revised version.. The term “C/N stoichiometry” (or any elements, for that matter) recently has been increasingly used in the articles. See, for example, Leifeld et al. (2020), Liu & Wang (2020), and others.

References

 Leifeld, J., Klein, K., & Wüst-Galley, C. (2020). Soil organic matter stoichiometry as indicator for peatland degradation. Scientific reports, 10(1), 7634. https://doi.org/10.1038/s41598-020-64275-y

Liu, R., & Wang, D. (2020). Soil C, N, P and K stoichiometry affected by vegetation restoration patterns in the alpine region of the Loess Plateau, Northwest China. PloS one, 15(11), e0241859. https://doi.org/10.1371/journal.pone.0241859

Point 26

Line 182 – 183 I don’t understand the connection to their preferences for recalcitrant plant matter?? Any reference?

Response 26

Here we tried to explain our finding of increased abundance of Acidobacteria in the subsoil. In the lower subsoil horizons the organic matter input results from the root litter and exudates and necromass of biota, with some low molecular mass substances leached from the above horizons, there was an increased C/N ratio in SOM, suggesting its increased recalcitrance to decomposition and bacterial utilization. We consider it importance to provide some explanation steering researchers interested in Acidobacteria’s metabolic potential  in this direction.

Point 27

Line 188 – usually other studies set up a threshold to assess rare communities. Perhaps, this could help to make this study more robust.

Response 27

Unfortunately, we do not see how removing some data will make the study more robust. The concept of robustness of a microbiome study can briefly be defined as based on the use of different methods on the same experimental system (Schloss, 2018). So how cutting-off some information may be helpful in this respect? We see it as an opportunity to give the information for other researchers in the field, seeking to compare their results with others for better understanding the microbial assemblages and how they interact with their environments. Personally, we often cannot find the basic data information we want, which may be really frustrating.  

Reference

Schloss P. D. (2018). Identifying and Overcoming Threats to Reproducibility, Replicability, Robustness, and Generalizability in Microbiome Research. mBio, 9(3), e00525-18. https://doi.org/10.1128/mBio.00525-18

Point 28

Line 189 – section 3.2 is more about Fungi abundance and distribution than diversity. Most usually diversity refers to diversity indices.

Response 28

We found out that many researchers, strictly speaking, include different meanings into the term ”diversity”. Diversity, as biodiversity, usually includes both, i.e. the composition (richness) and structure (abundance) of living organisms, comprising a community/assemblage/guild. That is exactly what this subsection is about.

Point 29

Line 194 – check all informal language along with the manuscript – e.g. quite a lot

Response 29

We removed or appropriately substituted the word “quite” everywhere in the manuscript; the changes are highlighted in turquoise in the revised version.

Point 30

Line 199 – Ascomycota vs Basidiomycota both with cursive??

Response 30

Yes, both should be cursive. We corrected the font appropriately, and highlighted the change in the revised version.

Discussion

Point 31

Discussion can be shortened. Most important to include is the contribution of this study to general knowledge. What the purpose of this kind of studies and what would be the difference with before and after your study.

Response 31

Point 32

Line 284 – 287  other forest but in the same soil type/climatic conditions? I think is not appropriate to assume that the discrepancy is due to bioinformatic tools.

Response 32

 Naturally, soil and climatic conditions can be different, but not drastically for similar types of biotopes/biomes. Therefore at the higher taxonomic levels one can reasonably expect some similarity in undisturbed habitats.

Point 33

Line 296 – 298 then you should include unclassified OTUs in diversity indices calculations to understand microbiome diversity as well as archaea group. Other options are the use sequence variants analysis instead of OTUs.

Response 33

Naturally, we included all OTUs, even the ones of just unclassified Bacteria, into the calculation of the α-biodiversity indices. Nowhere in the manuscript had we stated that we removed such OTUs from the analyses. Unfortunately, we could not identify the statements that could have given you such idea and hence could not make any changes. As for Archaea, we did not see any relevance of using the data about them obtained with universal bacterial primers.

Point 34

Line 310 – 320 maybe some references to pH preferences should be included

Response 34

We added the reference (31.    Eichorst, S.A.; Trojan, D.; Roux, S.; Herbold, C.; Rattei, T.; Woebken, D. Genomic insights into the Acidobacteria reveal strategies for their success in terrestrial environments. Environ. Microbiol. 2018,  20, 1041-1063. https://doi.org/10.1111/1462-2920.14043); it is highlighted in turquoise in the revised version.

Point 35

Line 371 – OTUs isn't a diversity index but ‘Observed OTUs’.

Response 35

Corrected by adding “observed”; the change is highlighted in turquoise in the revised version.

Point 36

Line 398 – writing sounds weird

Response 36

We corrected the weird wording by substituting “The latter suggests that diverse soil environment there” with “This finding suggests that diverse soil conditions in the topsoil horizons”;  the change is highlighted in turquoise in the revised version.

Point 37

Line 398 – 404 reference?

Response 37

We are afraid, we cannot see the pertinence of providing such reference in support of what is a very broad and general physiological concept.

Point 38

Line 428 – Error!

Response 38

The “Error!” message was substituted with a correct reference number; the change is highlighted in turquoise in the revised version.

Point 39

Line 431-435 Interesting discussion. Check how this relates to phenosoils vs genosoils https://www.sciencedirect.com/science/article/pii/S0016706117322401

Response 39

As far as we were able to understand (we are not specialists in soil mapping and pedogenesis), the article deals with agricultural (viticultural soils). Understandably, for the area of 220 sq.km sampling the same layer throughout (0-10  and 40-50 cm depth), irrespective of the genesis of its properties under the variability of soil forming factors in the studied valley, was the only feasible sampling option. However, we believe that different genesis of these layers that might be the case in some sites, contribute to the found differences between genoforms and phenoforms. But for the mapping purposes to get some kind of scaffolding for evaluating changes and presenting them to decision-makers the approach may be OK. (Sorry for this brief reply as we wanted to respect the deadline for submitting the revised version of our manuscript).    

Conclusion

Point 40

See contribution from these studies in undisturbed soils to assess losses in soil biodiversity by climate change and other human activities…

Response 40

To accommodate this, we inserted the phrase “for revealing changes that are likely to occur in the future due to climate warming, human activities, different land use or other factors”; the insertion is highlighted in turquoise in the revised version.

Point 41

Line 444 – Error!

Response 41

The “Error!” message was substituted with a correct reference number; the change is highlighted in turquoise in the revised version.

Point 42

Line 449 – misspelling

Response 42

We checked and dare say we do not see any misspelling there. As you know, “horizontation” means the 1) the structure of a natural body in a sequence of horizons, and 2) the methodology to sample such object and interpret the results. The term is used in the appropriate context in studied related to soil science. Here are some recent examples.  “For that purpose, other soil properties such as texture, water‐holding capacity, permeability, horizontation, or the extension of the modeled depth to the depth of bedrock may be of interest” (Rentschler et al., 2020, https://acsess.onlinelibrary.wiley.com/doi/full/10.1002/vzj2.20062).

Another example: “Differences in Fe stocks, vertical distribution of Fed, Feo and Fep and horizontation of equally old soils indicated an influence of microclimate on pedogenesis determined by slope aspect.” (Pollmann et al., 2020, abstract, https://www.sciencedirect.com/science/article/abs/pii/S2352009419302457)

Point 43

English and grammar should be improved throughout the entire manuscript. e.g. formal scientific language.

Response 43

We removed or appropriately substituted the word “quite” everywhere in the manuscript; the changes are highlighted in turquoise in the revised version.

Reviewer 2 Report

Missing of a clear-cut hypothesis is a major drawback of the manuscript. Data analysis is well performed. However, the arrangement of the manuscript was quite casual. The manuscript can be resubmitted again after a rigorous revision. The authors should spend more time in the discussion section. I think that the authors need to critically evaluate the manuscript before resubmitting again. Although the list is too long; below, I have summarized some points that need to be addressed.

  1. An abstract should contain a preliminary background of the proposed work with a clear-cut hypothesis, brief materials and methods, quantitative results and a solid conclusion. However, a systematic abstract is missing in the present manuscript. Result part is completely vague. Without any quantitative experimental data, it is not possible to make any statement. I request the authors to reorganize the abstract, please.
  2. The introduction part should rewrite again. Please add data to attract the attention of the reader about the topic. Aims of the study are provided. However, a clear-cut hypothesis is missing.
  3. From where meteorological data was taken. Please provide a reference. If data was recorded by authors then give details about the procedure.
  4. at line 177 and other lines, authors have said that same correlation. It's a confusing term. The correlation should be either significant or non-significant positive or negative.  Please see that.
  5. It is not enough to report whether their results are in agreement or not with the previous study but they should indicate why. The mechanism behind the treatment should be explained and elucidated. Discussion cannot enough to support its conclusion.
  6. The discussion part is not technically sound. Authors make many vague statements, which needs to be clarified. Discussion is not enough to support its conclusion.
  7. Please give a conclusive conclusion with the future aspects rather than descriptive.

Author Response

Response to Reviewer 2

Point 1

An abstract should contain a preliminary background of the proposed work with a clear-cut hypothesis, brief materials and methods, quantitative results and a solid conclusion. However, a systematic abstract is missing in the present manuscript.

Response 1

Unfortunately, we cannot help but saying that, as far as the abstract genre is concerned, we provided enough background. As for the clear-cut hypothesis, it is a descriptional study which aim was to obtain the data that can be used later to ignite hypotheses. We believe the abstract of our manuscript provides brief materials and methods, quantitative results and a solid conclusion.

Point 2

Result part is completely vague.

Response 2

This conclusion strongly produces the impression of not being pertinent to our manuscript, as the Results section in our manuscripts is usually the one for which reviewers have not comments   or, if any, stating that it is clearly written. Of course, personal perceptions vary, and some points may seem vague, but saying that it is “completely vague” seems to be totally unjustified.

Point 3

Without any quantitative experimental data, it is not possible to make any statement.

Response 3

Unfortunately, once again we are not able to grasp the point and hence make any appropriate corrections to improve the manuscript. There was no experiment at all: the study was a descriptional one. Such studies, as you know, provide an informational basis to develop hypothesis and use them to design experimental research.

Point 4

I request the authors to reorganize the abstract, please.

Response 4

We are afraid we cannot do it as we cannot see the need to.

Point 5

The introduction part should rewrite again.

Response 5

We rearranged two paragraphs in the Introduction section, starting it with soil since we submitted the manuscript for Soil Systems, and putting “forests” somewhat below. Since we did not rewrite anything, we did not highlight this rearrangement; besides, we tried to keep the introduction brief and to the point, and the reference list short by not supporting widely acknowledged, as we see them, statements.

Point 6

Please add data to attract the attention of the reader about the topic.

Response 6

We believe we provided enough information to attract readers: since we could not find any data, we made the relevant statement in the version you reviewed. Please, do mind that most of the studies devoted to soil “profiles” mean incremental sampling  sampling with depth in centimeters (and not too deep, i.e. not in the subsoil), rather than following the sequence of soil genetic horizons.

Point 7

However, a clear-cut hypothesis is missing.

Response 7

Unfortunately, once again we are not able to grasp the point and hence make any appropriate corrections. There was no hypothesis at all, clear-cut or not, as the study was a descriptional one. Such studies, as you know, provide data basis to develop hypotheses and use them to design experimental research.

Point 8

From where meteorological data was taken. Please provide a reference.

Response 8

We added the information for the source as a footnote https://meteoinfo.ru/en/climate/monthly-climate-means-for-towns-of-russia-temperature-and-precipitation and highlighted it in turquoise.

Point 9

at line 177 and other lines, authors have said that same correlation. It's a confusing term. The correlation should be either significant or non-significant positive or negative.  Please see that.

Response 9

We added the word “pattern” into the indicated line. Statistical significance of correlation is given in the Figure captions. So we did not repeat the information in the text, speaking about correlation only when is was statistically significant. However, the “correlation pattern” is another matter as it helps to visualize taxa relationships with soil properties and taxa similarity in this respect.   

Point 10

It is not enough to report whether their results are in agreement or not with the previous study but they should indicate why.

Response 10

We thought it was obvious that if someone writes about “agreement”, it means that the findings obtained are similar to findings reported by the cited studies. All our statements like this are quite clear.

Point 11

The mechanism behind the treatment should be explained and elucidated.

Response 11

We are afraid that we should reiterate that there was no treatment as it was not an experimental study; therefore we can supply neither explanation, nor elucidation about/on this point.

Point 12

Discussion cannot enough to support its conclusion.

Response 12

In the Discussion section we tried to put our results in the framework of the current knowledge on the subject, and believe we did a decent job. Especially with a point about soil horizontation, which is of utmost importance both for experimental and meta-analytical soil microbiome research.

Point 13

The discussion part is not technically sound.

Response 13

Unfortunately, we could not see why you think so, as in general it follows the lines/aspects presented in other papers on the topic.

Point 14

Please give a conclusive conclusion with the future aspects rather than descriptive

Response 14

We are afraid, that, since the study was a descriptive one, we cannot possibly be more conclusive in this respect.

Round 2

Reviewer 2 Report

Dear Authors

I have found the article satisfactory for publication. All the corrections suggested are incorporated.